# Complement Binding Anti-HLA Antibodies and the Survival of Kidney Transplantation

**DOI:** 10.3390/jcm12062335

**Published:** 2023-03-17

**Authors:** Claudia M. Muñoz-Herrera, Juan Francisco Gutiérrez-Bautista, Miguel Ángel López-Nevot

**Affiliations:** 1Departamento de Bioquímica, Biología Molecular e Inmunología III, University of Granada, 18010 Granada, Spain; 2Programa de Doctorado en Biomedicina, University of Granada, 18010 Granada, Spain; 3Servicio de Análisis Clínicos e Inmunología, Hospital Universitario Virgen de las Nieves, 18014 Granada, Spain; 4Clínica Imbanaco Grupo Quirónsalud, Laboratorio Clínico, Patología y Servicio de Transfusión, Laboratorio de Inmunogenética, 760042 Cali, Colombia; 5Instituto de Investigación Biosanitaria de Granada (ibs.GRANADA), 18012 Granada, Spain

**Keywords:** AMR, antibody-mediated rejection, DSA, donor-specific antibodies, dnDSA, de novo donor-specific antibodies, complement-fixing DSA, renal transplant, kidney graft

## Abstract

Background: Antibody-mediated rejection (AMR) is one of the most important challenges in the context of renal transplantation, because the binding of de novo donor-specific antibodies (dnDSA) to the kidney graft triggers the activation of the complement, which in turn leads to loss of transplant. In this context, the objective of this study was to evaluate the association between complement-fixing dnDSA antibodies and graft loss as well as the possible association between non-complement-fixing antibodies and transplanted organ survival in kidney transplant recipients. Methods: Our study included a cohort of 245 transplant patients over a 5-year period at Virgen de las Nieves University Hospital (HUVN) in Granada, Spain. Results: dnDSA was observed in 26 patients. Of these patients, 17 had non-complement-fixing dnDSA and 9 had complement-fixing dnDSA. Conclusions: Our study demonstrated a significant association between the frequency of rejection and renal graft loss and the presence of C1q-binding dnDSA. Our results show the importance of the individualization of dnDSA, classifying them according to their ability to activate the complement, and suggest that the detection of complement-binding capacity by dnDSA could be used as a prognostic marker to predict AMR outcome and graft survival in kidney transplant patients who develop dnDSA.

## 1. Introduction

Organ transplantation is defined as the replacement of dysfunctional organs with healthy tissues or other organs. Renal transplantation remains the method of choice for patients with end-stage renal disease, because it offers better survival, better quality of life, and reduced costs compared to dialysis [1,2,3]. Human leukocyte antigens (HLA) are highly polymorphic molecules that must be considered for transplant matching, as they can induce a strong immune response with detrimental effects on graft survival [4]. A higher degree of HLA compatibility between the donor and recipient is better for graft survival, with a lower probability of acute rejection and sensitization of the recipient to donor HLA molecules. Therefore, better results are obtained when transplanted with an HLA-identical donor (sibling) compared to haploidentical or less-compatible donors, such as a deceased donor [5,6,7].

The presence of donor-specific antibodies (DSA) is a barrier to consider in renal transplantation. Sensitization against non-self HLA molecules occurs as a result of previous sensitizing events, such as pregnancy, blood transfusions, infections, and transplants. The strongest and longest-lasting degrees of immunization are related to previous transplants [8]. Therefore, it is necessary to determine anti-HLA antibodies before transplantation in order to avoid rejection of the transplanted organ.

Rejection can be classified according to when it occurs as hyperacute rejection or as acute or chronic rejection, and it can be classified by the mechanism that causes it as cell-mediated rejection, antibody-mediated rejection, or mixed rejection [9]. Despite the immunosuppressive therapy that accompanies transplantation, rejection still occurs in approximately 15% of patients. However, evidence has shown that when rejection episodes are diagnosed early, modulating immunosuppression therapy, either by increasing its dose or by providing additional drugs, usually yields successful results in preventing graft loss. Treatment depends upon the type and severity of the rejection [9,10].

Antibody-mediated rejection (AMR) occurs when the immune system recognizes the differences between self and foreign, primarily through the recognition of the donor’s HLA molecules. AMR is currently one of the most important challenges in the context of transplantation [11,12,13]. The antibody-mediated immune response requires contact between an antigen and a specific antibody. This union will activate different effector mechanisms for the elimination of the foreign antigen. One of the main effector mechanisms is the classical complement pathway. The activation of the complement involves the cascading proteolysis of different serum proteins that bind to the antigen–antibody complex so that the membrane attack is carried out with the consequent elimination of the foreign [14,15]. The binding of the DSA to the renal graft triggers the activation of the complement cascade, which leads to organ damage and loss of the transplant. However, the presence of DSA, which does not have complement-binding capacity, may not lead to an immune response that induces rejection of the transplanted organ [16].

The objective of this study was to evaluate the possible involvement of complement-fixing DSAs in renal graft loss and the possible association between non-complement-fixing antibodies and renal graft survival.

## 2. Materials and Methods

### 2.1. Study Population

This study included 245 renal transplant patients between 2014 and 2018 at the Virgen de las Nieves University Hospital (HUVN) in Granada, Spain. The mean age of the patients was 54 years; 71% were men (n = 174), and 29% were women (n = 71) (the age of our recipients was similar to the range reported in most cases of ESRD in the general population [17]). The mean donor age was 50 years. The characteristics prior to renal transplantation are summarized in Table 1.

Patients who had the following data available were included in our study: serum creatinine (SCR) values after transplantation, anti-HLA antibody analysis, and biopsy results in cases of rejection. In addition, the patients were required to have a serum sample for complement-binding antibody analysis. DSA monitoring and biopsies were conducted on all the patients by protocol after the transplant. In the cases of the presence of dnDSA and if the biopsy showed changes that suggested rejection, the immunosuppression was modulated until the symptoms were controlled and the DSA was not detected in some cases.

All patient samples were collected according to local medical ethics regulations after informed consent was obtained from the subjects, their legal representatives, or both, according to the Declaration of Helsinki. Studies involving human participants were reviewed and approved by the ethics committee of the Portal de Ética de la Investigación Biomédica de Andalucía (PIEBA) of the Andalusian government (Code:0766-N-20). All patients provided consent for the publication of the study.

### 2.2. Determination of DSA Complement Binding Capacity: C1q Technique

In order to determine the complement binding capacity of anti-HLA antibodies, serum samples with DSA were used. The serum samples were duly labeled with patient data and the date of extraction and were stored at −80 °C.

Complement binding capacity was determined using the commercial C1qScreen™ kit from One Lambda (One Lambda, Canoga Park, CA, USA). The One Lambda HLA LABScreen™ and C1qScreen™ HLA assays allow for the detection and identification of immunoglobulin G (IgG) in serum samples directed against complement-binding HLA class I and II molecules. This method is based on the specificity of antigen–antibody (Ag-Ac) binding recognized by the C1q molecule. This reaction is detected by an antibody labeled with a fluorophore, the signal of which is analyzed using a fluoroanalyzer. Subsequently, the data obtained were analyzed using HLA Fusion™ software version 4.3 (One Lambda), which allows for the specificity of the antibody to be assigned by comparing the fluorescence of each sphere with the assigned antigen [18,19]. All of the tests were performed according to the manufacturers’ instructions.

Antibody positivity assignment was performed based on a mean fluorescence intensity (MFI) value of 1500 according to HUVN clinical protocols. The DSA antibody assay was tested for HLA-A, -B, -C, -DR, and -DQ.

### 2.3. Statistical Analysis

Statistical analysis of the data and presentation of the results were performed using IBM-SPSS V.21.0 (Statistical Package for the Social Sciences) version 21.0. (IBM Corp., Armonk, NY, USA).

Descriptive statistics: The profile of the study population was described. The results of categorical variables were expressed as percentages, and those of quantitative variables were expressed as mean and standard deviation, including confidence intervals (95% CI) and range (minimum and maximum values).

Bivariate analysis: The X2 test or two-tailed Fisher’s exact test was used, when necessary, with contingency tables in order to compare the proportions between groups. In order to analyze the differences between the mean values of quantitative variables between the two groups, Student’s t-test for independent samples (after analysis of equality of variance using Levene’s test) or its nonparametric equivalent, the Mann–Whitney U test, was applied.

Survival analysis: Survival analysis was performed using the Kaplan–Meier method. Statistical significance between the survival times was determined using the log-rank test. Differences were considered statistically significant at a corrected *p*-value < 0.05.

## 3. Results

### 3.1. Descriptive Analysis of the Population and Characteristics Prior to Renal Transplantation

The following pre-transplant clinical variables were analyzed in the recipients: number of HLA incompatibilities (for HLA-A, -B, and -DR loci), percentage of sensitization (cPRA), and the presence of pre-transplant DSA and its complement binding capacity (Table 1).
jcm-12-02335-t001_Table 1Table 1Description of variables pre-renal transplantation.

FrequencyPercentage (%)Total Incompatibilities031.2120.8252.032811.446124.958936.365723.3Total245100

FrequencyPercentage (%)cPRA%Negative19278.41–50%3112.751–94%124.995–100%104.1Total245100

FrequencyPercentage (%)DSA pre-transplantNO24098YES52Total245100

FrequencyPercentage (%)DSA pre-transplant complement fixersNO5100YES00Total5100Total incompatibilities: The number of incompatibilities for HLA-A, HLA-B, and HLA-DR loci were considered (Human leukocyte antigens (HLA)). cPRA%: Percentage of calculated PRA (presence/absence of anti-HLA antibodies and their percentage of sensitization). DSA: donor-specific antibodies identified at the time of transplantation and pre-transplantation with complement binding capacity (binding to C1q).

Five patients had DSA at the time of transplantation. Complement-dependent cytotoxicity crossmatch (CDC) results were negative in all patients; therefore, they were used for transplantation. The C1q-binding capacity of DSA was determined, and the results were negative. In DSA analysis, three patients were found to have antibodies against the HLA-A locus, one against the HLA-DR and -DQ loci, and one patient against the HLA-DQ locus.

During our study, these five patients did not present with rejection events or renal graft loss. Follow-up was performed between 6 months and 4 years post transplantation (mean 2 years). In two patients whose presence of DSA persisted post-transplantation, the MFI values of DSA were <5000. The results did not show a statistically significant *p*-value for rejection events or renal graft loss due to the presence of pre-transplant DSA.

### 3.2. Analysis of Post-Renal Transplant Characteristics

Analysis of the post-renal transplantation characteristics was performed by identifying the three populations in our study. The populations were classified according to the presence or absence of DSA and the complement binding capacity (Table 2). A total of 219 patients (89.4%) did not develop DSA, and 26 presented with de novo DSA (dnDSA). Of the latter, 17 patients (6.9%) had dnDSA without complement fixation and nine patients (3.7%) had dnDSA with complement fixation. Table 2 summarizes the demographic data and immunological characteristics of the three populations identified. Among the recipients who developed dnDSA, the majority (42.3%, n = 11) were against HLA-II, 34.6% (n = 9) were against HLA-I, and 23.1% (n = 8) were against both HLA-I and HLA-II.

### 3.3. Rejection Events

In our cohort, 23 patients (9.4%) experienced rejection. These were classified according to the cause of origin as cellular rejection, humoral rejection, mixed rejection, or non-immunological causes of rejection, according to biopsy results. Non-immunological graft failures occurred in 10 patients (4.1%) and were associated with surgical problems (graft laceration), infections, lymphocele, or immunosuppressant toxicity. The results were poor renal function, hemorrhagic shock, necrosis, and graft loss. Nine patients lost their grafts during the first month. The other patient lost the graft at six months due to a lymphocele. None of the patients had DSA at the time of organ rejection.

We found that only 10 patients (38.5%) had rejection events with dnDSA, whereas the remaining 13 (61.5%) were dnDSA-negative. Subsequently, we analyzed the complement-binding capacity of dnDSA and its association with rejection events in 10 patients. We found that seven patients had dnDSA with a complementary binding capacity. Three patients developed humoral-type events, and four had mixed-type events. In contrast, of the patients with dnDSA without complement binding capacity, two had humoral-type rejection events and one had a mixed-type rejection event. Our analysis showed a statistically significant association between the presence of dnDSA and the complement binding capacity and rejection events (*p* = 0.009) (Figure 1). In addition, we found two patients with dnDSA with complement-binding capacity who did not present with any rejection event at the time of this study.

### 3.4. Renal Graft Survival Associated with the Presence of dnDSA

The median follow-up time after renal transplantation in patients without dnDSA was 17 months (range, 0–61 months). For patients with dnDSA with C1q binding capacity, it was 26 months (0–49 months), and for those with dnDSA without complement fixation, it was 17 months (0–63 months). The results showed that patients with dnDSA had shorter survival to renal transplantation than those without dnDSA (*p* = 0.017) (Figure 2).

When determining renal transplant survival in relation to C1q binding capacity, it was observed that patients with complement-fixing dnDSA were associated with lower renal graft survival than those with non-complement fixers (*p* = 0.009)

Finally, transplant survival was evaluated in the three populations in our study. The results showed that patients with dnDSA that did not bind to C1q and those who did not develop dnDSA showed similar graft survival. However, individuals in the latter group presented graft loss due to non-immunological causes, which explains the differences between these two populations in the graph. Finally, the complement-fixing dnDSA group presented the lowest survival with the highest association with the risk of graft loss (*p* < 0.001) (Figure 3). 

### 3.5. Renal Graft Loss

Renal graft loss occurred in 18 (7.3%) patients. Ten patients lost the transplant due to non-immunological causes, three patients due to cellular rejection without the presence of dnDSA at the time of graft loss, and five patients due to immunological rejection associated with the presence of dnDSA (one patient due to humoral rejection and four due to mixed rejection). We analyzed whether the immunological events that led to transplant loss in the latter group were related to the presence of complement-binding antibodies. The results suggested the presence of C1q-binding antibodies in all cases, with a statistically significant association between the presence of complement-fixing antibodies and renal graft loss (*p* = 0.002) (Figure 4).

### 3.6. Relationship between MFI Values of dnDSA and Its C1q Binding Capacity

We performed statistical analysis in order to determine the possible association between the MFI values of post-transplantation dnDSA and their complement-binding capacity. Based on the classification model proposed by Zecher et al. [20], three categories were established to classify MFI values: MFI less than 5000, MFI between 5001–9999, and MFI greater than 10,000. We found a statistically significant association between MFI and complement fixation capacity (*p* = 0.033). Therefore, we proceeded to determine where this difference lies; in order to accomplish this, the same variables were compared in 2 × 2 tables, showing a statistically significant association between MFI values less than 5000 and those above 10,000 (*p* = 0.017); the rest of the associations were not statistically significant (Figure 5). 

## 4. Discussion

AMR is one of the primary causes of renal graft loss. The main risk factor is incompatibility with donor HLA molecules owing to the development of antibodies against allogeneic HLA. Finding a good match is essential; however, given the high polymorphism of HLA molecules, it is becoming increasingly difficult to find an optimal donor. In addition, renal patients are at a high risk of sensitization events due to receiving transfusions, leading to the development of anti-HLA antibodies, which represent a barrier for the patient to be transplanted [21].

We retrospectively reviewed a cohort of 245 patients who underwent renal transplantation into the HUVN. We found that five patients showed evidence of pre-transplant DSA. However, at the time of this study, none of the patients had episodes of AMR or renal graft loss. The determination of complement fixation properties by DSA could contribute to risk measurement, as has been proposed by different authors [20,22,23,24]. The results of these patients demonstrated that DSA was not able to activate complement. Therefore, these patients were less likely to develop rejection and had a positive predictive value for graft survival [20,25]. Different studies have discussed whether the presence of pre-transplant DSA is a contraindication for transplantation [22,23]. However, despite the lack of information on the role that DSA may play in pre-transplantation and renal graft survival, renal transplantation in the presence of DSA is still under debate, especially when they do not fix complement [24]. However, it is possible that the induction into an immunosuppressive regimen that these patients received, resulting in continued immunosuppression, allowed for the elimination of alloreactive T- and B-cell clones and the disappearance of pre-existing DSA. These patients were comparable with those who did not develop dnDSA. It should be noted that continuous follow-up of these patients is necessary for the early detection of DSA reappearance and its ability to activate the classical complement pathway [9,26]. Contrary to our findings, other authors have argued that the presence of DSA before transplantation is a risk factor for the development of acute AMR, poor renal function outcomes, and graft loss. Therefore, it is essential that at-risk patients undergo a good evaluation along with stricter follow-up after transplantation [21,27,28,29].

Our study demonstrated a significant association between the frequency of rejection events and the presence of C1q-binding dnDSA. The presence of dnDSA, which did not bind to the complement, was associated with fewer rejection events. Therefore, we found that C1q determination provides added value in the early detection of possible renal graft injury mediated by complement-binding dnDSA, which may lead to graft loss if not treated promptly. All of these associations have been widely discussed in multiple studies [30,31,32,33], confirming the importance of dnDSA categorization.

The renal graft survival assessed in our population showed similar results to the transplant survival reported for patients who developed dnDSA in different studies [27,30,34]. In our cohort, renal survival rates at five years after transplantation were significantly decreased in patients who developed dnDSA. However, when classified according to their C1q-binding capacity, graft survival remained similar for patients without dnDSA and those with dnDSA without complement-binding capacity. In contrast, the presence of dnDSA with C1q-binding capacity was indicative of an increased risk of renal graft loss. These results correspond with those published by Loupy et al., who found an association between the risk of transplant loss and the presence of antibodies that activate the classical complement pathway [12]. Research by Tyan et al. supports this association and estimates that the presence of dnDSA can lead to loss within 1 to 2.5 years [27].

Among the recipients who developed dnDSA, 42.3% developed dnDSA only to HLA-II, 34.6% developed dnDSA only to HLA class I, and 23.1% developed antibodies against both HLA-I and HLA-II. Our data are similar to the percentages of dnDSA reported in other studies [16]. However, the association between MFI values and complement binding capacity by dnDSA agrees with that published by different authors [35,36]. This suggests that higher MFI values reflect elevated antibody titers and higher C1q binding capacity and are generally associated with an increased risk of developing AMR. Despite this association, in our study, we found that patients with C1q-binding of dnDSA with MFI greater than 10,000 had no rejection events. Similarly, we found that patients with non-C1q-binding dnDSA with MFI less than 5000 developed rejection events without graft loss. Our findings for these cases are similar to those published in several studies, which found that the association with MFI values was not equal for all dnDSA, and some could bind to C1q regardless of their MFI value [27,32,37].

Another important finding of our study was that the majority of dnDSA with C1q-binding capacity were directed against the HLA-DQ locus (7 of 9 patients). These were associated with rejection events regardless of MFI value. No associations were found that demonstrated a significant effect from the presence of antibodies against HLA-DQ and the capacity to bind complement; however, a trend was observed that had a value close to significance (*p* = 0.097). These findings suggest that the presence of antibodies against HLA-DQ leads to a higher risk of complement activation even when MFI values are low, as reported in other investigations [33,38,39]. Different authors have published their observations on the presence of antibodies against HLA-DQ early in renal transplant rejection and the consequent development of ABMR, transplant glomerulopathy, and its implication in graft survival [40,41,42]. Given these findings, organ allocation policies, especially those of Eurotrasplante and the Organ Procurement and Transplant Network (OPTN), have widely discussed the inclusion of the HLA-DQ match as a matching strategy in order to decrease the risk of developing de novo DSA and graft rejection. For its part, it is expected that with the reduction of the mismatch in the HLA-DQ molecule, the probability of rejection events will be significantly lower, which translates into benefits for the patient by reducing exposure to rescue immunosuppression therapies, which affect morbidity and mortality rates [43]. Based on our results, it is important to highlight the importance of searching for compatibility in the HLA-DQ locus in order to reduce the risk represented by the immunogenicity of this antigen.

Progress in the development of more sensitive techniques for the identification of dnDSA has allowed for the monitoring of transplant patients in a less-invasive way. However, the classification of dnDSA according to their complement fixation capacity could be an important step in post-transplant follow-up. Given the utility of this technique, it could predict the pathogenicity of dnDSA and allow the risk stratification of patients. It is important to recognize that much remains to be learned about the pathogenesis of dnDSA in post-transplant follow-up [30]. However, follow-up protocols could include the analysis of complement-fixing antibodies to more accurately classify these patients, especially those with stable renal function, before performing a graft biopsy [21,34].

The findings of this study support the hypothesis of the importance of the individualization of dnDSA, classifying them according to their ability to activate the classical complement pathway, risk of developing AMR events, and inducing graft loss. Therefore, the possibility of evaluating C1q binding in sensitized patients who are candidates for kidney transplantation, especially hyperimmunized patients, is a promising alternative. The studies already published, together with the data presented here, open up the possibility of being able to accept organs that would be considered “incompatible”, as a strategy to reduce mortality rates on the waiting list, and constitutes an interesting future perspective to extend this study [20,38,44].

## 5. Conclusions

Our results suggest that the detection of complement-binding capacity by dnDSA could be an additional prognostic marker to predict AMR outcome and graft survival in dnDSA-positive kidney transplant patients. Although our results are consistent with those previously published, the low number of renal rejection cases may be a limiting factor for this study.

## Figures and Tables

**Figure 1 jcm-12-02335-f001:**
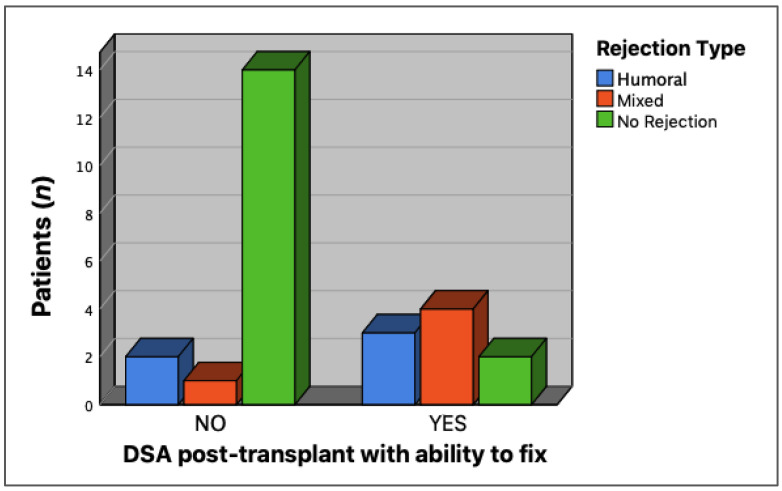
Association between the presence of complement-binding dnDSA and rejection events. A higher number of rejection events was observed in patients who developed C1q-binding dnDSA, with respect to those who did not bind complement (*p* = 0.009).

**Figure 2 jcm-12-02335-f002:**
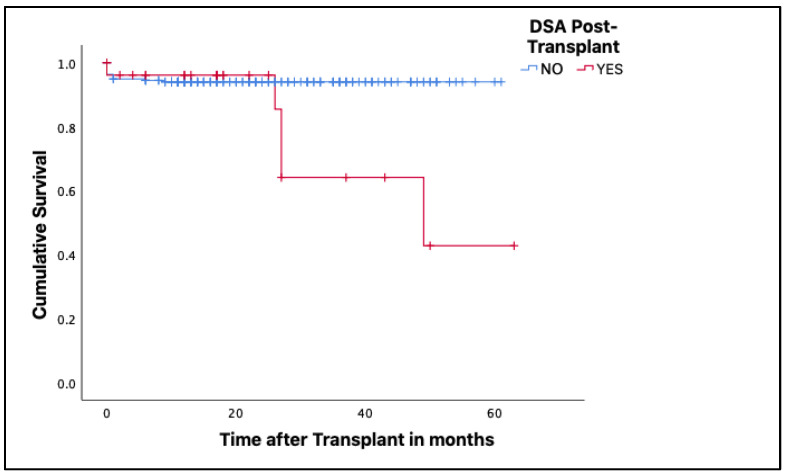
Kaplan–Meier curves to determine renal graft survival, according to the presence or absence of dnDSA. Lower renal graft survival is observed in patients who develop dnDSA.

**Figure 3 jcm-12-02335-f003:**
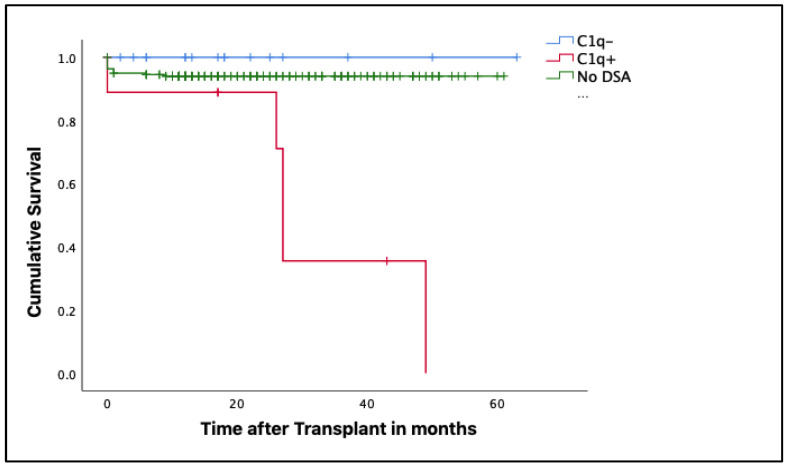
Kaplan–Meier curves to determine renal graft survival according to C1q-binding capacity by dnDSAs. Lower renal graft survival is observed in patients who develop complement-binding dnDSAs, whereas survival is similar for the other two populations.

**Figure 4 jcm-12-02335-f004:**
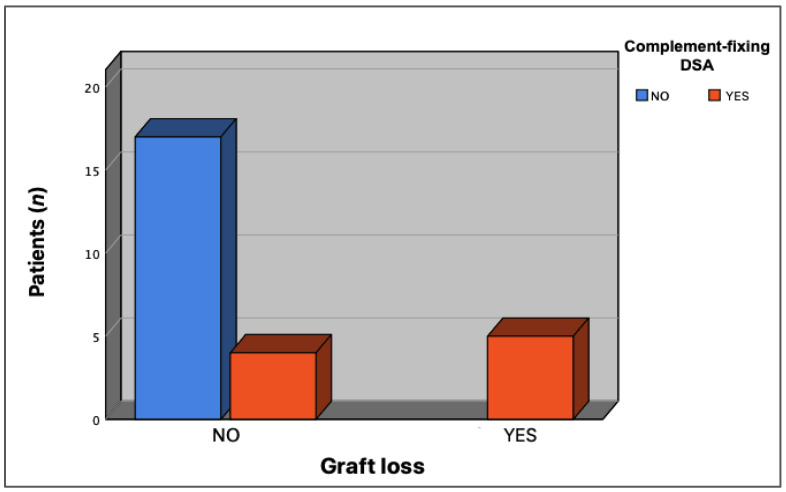
Association between the presence of complement-binding dnDSA and renal graft loss. Renal graft loss was observed in patients who developed C1q-binding DSA, with respect to those who did not (*p* = 0.002).

**Figure 5 jcm-12-02335-f005:**
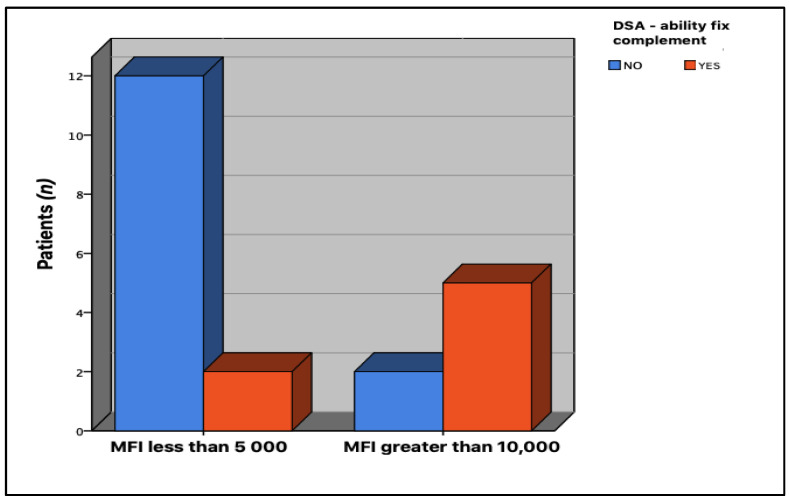
Association between MFI values and C1q binding capacity. Non-binding dnDSA comprise MFI values below 5000, whereas C1q-binding dnDSA are more related to MFI values above 10,000 (*p* = 0.017).

**Table 2 jcm-12-02335-t002:** Characteristics according to the presence or absence of DSA and its binding capacity to C1q.

Features	Total Patients	No DSA	DSA	
			Non-Complement Fixers	Complement Fixers
Recipient	245	219	17	9
Age	53.9 ± 12.4	54.2 ± 12.4	55.0 ± 11.9	46.1 ± 10.9
Sex	174 men	158 men	11 men	5 men
	71 women	61 women	6 women	4 women
Donors				
Age *	50.4 ± 14.7	51.2 ± 14.5	43.3 ± 16.7	46.0 ± 13.3
Sex *	166 men	147 men	12 men	7 men
	68 women	61 women	5 women	2 women
Type of Donors	233 deceased	207 deceased	17 deceased	9 deceased
	12 alive	12 alive	0 alive	0 alive
Immunological data				
HLA Incompatibilities	4.6 ± 1.1	4.6 ± 1.1	4.5 ± 1.5	4.5 ± 1.1
Blood Group Recipient				
A	105	98	7	0
B	30	30	0	0
AB	6	5	0	1
0	104	86	10	8
dnDSA				
HLA class I	9	-	8	1
HLA class II	11	-	7	4
HLA class I and II	6	-	2	4

Summary of the three populations identified based on the presence or absence of DSA and their C1q-binding capacity. dnDSA: de novo DSA. *: incomplete data.

## Data Availability

The raw data supporting the conclusions of this article will be made available by the authors without undue reservation.

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
