# Peer review of "Complement Binding Anti-HLA Antibodies and the Survival of Kidney Transplantation"

_jcm, 2023, doi:10.3390/jcm12062335_

Round 1

Reviewer 1 Report

This single center studies of the role of assessment of complement fixing DSA in kidney transplantation. Numbers are limit but acceptable for single center.

Abstract: The conclusions should be expanded to include outcome with C1q + dnDSA (background can be shortened.

Materials and Methods:   Was DSA monitoring done per schedule, for cause, or just at time of biopsy?

How were serquential positive DSA handled

Were biopsies by protocol and for cause or just for cause?

Results:

Page 4  line 166 – would recommend adding  “At the time of our  During our study, these five patients”

Page 6 line  194,195 – the term “Non-immunologic rejection events” do not seem to be rejection in conventional sense – would suggest using “Non-immunologic graft failures”

Page 6 Fig 1 legend “fix”  versus bind complement

Page 7 line 295 3 patients had cellular rejection but no cellular rejection

Discussion:

Page 9 “Although a posteriori was performed” phrase not needed

Page 9 line 390 “is still under debate, even when those antibodies are not complement fixators”

I would say it is under debate especially when they do not fix complement

Page 10 line 423 – need to specify 42.3 developed dnDSA only to HLAII, 34.6 developed dnDSA only to HLAI

Page 10 line 446 the information discussed in this paragraph was not included in the results

Author Response

"Consulte el archivo adjunto."

Reviewer 2 Report

The authors in this paper performed an analysis of their renal transplant recipients with regards to complement fixation of dnDSA detected. Overall, the analysis is sound and their  conclusions supported by the data. Though single center data, these data are sufficiently hypothesis generating so as to be of interest to a scientific community. 

Overall, the analysis is sound and well presented. I actually have no substantial concerns with the data as presented. 

In the abstract, I would include more information about how complement binding capcity of dnDSA influenced important transplant outcomes. For example, would include data regarding how complement binding capacity of dnDSA is associated with graft survival. 

There is a broad literature about the importance of HLA-DQ mismatch and subsequent development of dnDSA (see PMID: 34246656). This should be included in the discussion of this manuscript. 

Author Response

"Consulte el archivo adjunto."
